# The effectiveness of problem based learning in improving critical thinking, problem-solving and self-directed learning in first-year medical students: A meta-analysis

Ida Bagus Amertha Putra Manuaba[1,2‡], Yi -No[3], Chien-Chih Wu[3,4]*

1 College of Medicine, Taipei Medical University, Taipei, Taiwan, 2 Medical and Health Education Development, Faculty of Medicine, Udayana University, Bali, Indonesia, 3 Department of Education and Humanities in Medicine, School of Medicine, College of Medicine, Taipei Medical University, Taipei, Taiwan, 4 Department of Urology, Taipei Medical University Hospital, Taipei, Taiwan

‡ IBAPM are sole first author to this work.
* Ccwu@tmu.edu.tw

## Abstract

### Background

The adaptation process for first-year medical students is an important problem because it significantly affects educational activities. The previous study showed that 63% of students had difficulties adapting to the learning process in their first year at medical school. Therefore, students need the most suitable learning style to support the educational process, such as Problem-based learning (PBL). This method can improve critical thinking skills, problem-solving and self-directed learning. Although PBL has been adopted in medical education, the effectiveness of PBL in first-year medical students is still not yet clear. The purpose of this meta-analysis is to verify whether the PBL approach has a positive effect in improving knowledge, problem-solving and self-directed learning in first-year medical students compared with the conventional method.

### Methods

We searched PubMed, ScienceDirect, Cochrane, and Google Scholar databases until June 5, 2021. Search terms included problem-based learning, effectiveness, effectivity, and medical student. We excluded studies with the final-year medical student populations. All analyses in our study were carried out using Review Manager version 5.3 (RevMan Cochrane, London, UK).

### Result

Seven eligible studies (622 patients) were included. The pooled analysis demonstrated no significant difference between PBL with conventional learning method in critical thinking/knowledge assessment (p = 0.29), problem-solving aspect (p = 0.47), and self-directed learning aspect (p = 0.34).

**Data Availability Statement:** All relevant data are within the manuscript and its Supporting information files.

**Funding:** The author(s) received no specific funding for this work.

**Competing interests:** The authors have declared that no competing interests exist.

## Conclusion

The present study concluded that the PBL approach in first-year medical students appeared to be ineffective in improving critical thinking/knowledge, problem-solving, and self-directed compared with the conventional teaching method.

## Introduction

The adaptation process for first-year medical students is an important problem because it is one of the factors that significantly affect educational outcomes [1]. Struggling can occur at any time, but first-year students are particularly susceptible as they adapt to new learning methods at university [2]. A study on the adaptation process of first-year medical students involving 200 participants showed that 63% of students had problems adapting to the learning process [3]. Consequently, students need to know the most suitable learning style to support the educational process. In addition, the appropriate learning approach can also help the adaptation process of first-year medical students and maximize their study outcomes. Therefore, educational institutions need to ensure that applied learning methods improve the learning atmosphere for first-year medical students [4].

Problem-based learning (PBL) encourages students to identify their knowledge and skills to achieve specific goals [5]. Many studies have evaluated the effectiveness of PBL in the medical curriculum and found that PBL can improve understanding, team performance, learning motivation, student satisfaction, and critical thinking [5, 6]. The PBL method not only helps students to understand in-depth, but it also encourages independent learning in students because they have to formulate their own learning goals after understanding PBL scenarios, solve their problems via literatures and internet, compare scenarios with theories from various sources and actively participate in group discussions [7]. PBL has three main learning objectives, namely (1) to apply deep content learning, (2) to apply problem analysis skills and develop solutions to solve problems, and (3) to apply self-directed learning as an approach to adapt learning styles [8]. Therefore, this teaching model has been highly praised in medical education courses in the past two decades [9]. In conventional lecture methods, students are passively exposed to the material and less likely to learn or apply concepts actively. Meanwhile, in PBL, students will learn actively using case-based peer-to-peer teaching, stimulating students to learn based on lecture materials and independent learning to solve cases under the guidance of a facilitator. The PBL approach aims to promote the integration of learned knowledge, rather than simply implanting knowledge and skills compared with the conventional teaching model [8] and also has been design to emphasizes active participation, problem-solving, and critical thinking skills compared to conventional medical education practices [6].

Several reports have showed the effectiveness of PBL for the first-year medical students in improving the final score with the help of map concept compared to PBL only group. The average score was improved significantly, namely 10.07±3.49 versus 5.97±2.09, p<0.001 [10]. Another study compared the final score between the PBL method and the conventional method accompanied by a workshop for first-year medical students. The final results were also statistically significant, namely 8.25±0.79 versus 5.46±0.96, p<0.01 [11]. However, due to the limitation of the studies, the effect of PBL for first-year medical students is yet to be concluded. Also, there is still no meta-analysis that evaluates this topic to date. Therefore, we conducted a systematic review and meta-analysis to verify whether the PBL approach has a positive effect in

improving knowledge/critical thinking, problem-solving and self-directed learning in first-year medical students compared with the conventional method.

## Methods

### Study design

A Meta-analysis was performed from March to June 2021 to assess the effectiveness of PBL in improving knowledge/critical thinking, problem-solving and self-directed learning in first-year medical students. To attain our goal, potentially relevant papers were identified and collected from PubMed, Cochrane, ScienceDirect, and Google Scholar to calculate the mean difference and 95% confidence interval (95%CI) using a random and fixed-effect model. We used meta-analysis protocols as a guide in our present study [12].

### Search strategy

We conducted a systematic search in PubMed, Cochrane, ScienceDirect and Google Scholar for search strategy up to June 5, 2021. The search strategy conformed to medical subjects heading (MeSH), involving the use of a combination of the following keywords: (Problem-based Learning [MeSH Major Topic]) AND (effectiveness OR effectivity AND medical student AND first-year). Language constraints were applied in our quest policy. We only used the bigger sample size analysis, which was up to date when we saw the same results in the experiments. We also scanned the possible papers of the appropriate or qualifying studies reference list by searching "Articles linked". Two independent inspectors found potentially vital records (I.B.A. P.M, Y.N). Disagreements between two independent researchers related to the article were settled by a debate and/or consultation with the senior investigator for finding the third opinion (C.C.W).

### Eligibility criteria and data extraction

The inclusion criteria for this study included: (1) research subjects were medical students at the first year (first or second semester), (2) study that evaluated the knowledge/critical thinking, problem-solving and self-directed learning of the student, (3) study that provided sufficient data for calculation of mean difference and 95%CI, p-value, and study heterogeneity. Meanwhile, the exclusion criteria were as follows: (1) studies with insufficient data, (2) samples size less than 50, (3) intervention duration less than one year, (4) review, letter to the editor, and comments articles. Data extraction was conducted by two authors (I.BA.P.M, Y.N). Both of those authors independently screened the collected article's title, abstract, and full text. Two reviewers extracted the data, which was then extracted to Google Spreadsheet by two reviewers (I.BA.P.M, Y.N). Information was derived from each article included in this study as follows: (1) first author's name and year of release, (2) age of the participant, (3) interventional and control method, (4) sample cases and control sizes, (5) country of study, (6) study program, (7) duration of PBL intervention, (8) score of PBL and control group. Two independent authors carried out data extraction to prevent human mistakes. If there were a disagreement, a discussion would be held to discuss the solution.

### Quality assessment

Two independent authors (I.BA.P.M, Y.N) assessed the quality of the studies to ensure each sample's validity and prevent the possible exaggeration of each study. The authors use major and minor criteria in assessing the risk of bias for quality assessment. There were four major and four minor criteria. The authors assigned 2 points each to the major criteria and 1 point

each to the minor criteria so that the total score would be 12 points. If the article got 9–12 points, then it assigned as "low-risk bias," if the article got 6–8 points, then it assigned as medium risk bias", and if the article got < 5 points, then it assigned as "high risk of bias". When there was a disagreement between the two authors, a discussion was held. If the conflict has not been settled, the two authors discuss it with the third author (C.C.W).

## Statistical analysis

Assessment of Methodological Quality of Individual Trials in each article was assessed at the risk of bias before enrolling in meta-analysis. The Z-test was used to assess the effectivity learning method from self-directed learning and its sub-group analysis, critical thinking/knowledge, and problem-solving. Forest plots defined the group measurement and impact estimate. Heterogeneity was provided by using several parameters that we provide, such as $Chi^2$, $Tau^2$, and $I^2$. In the beginning, Comprehensive Meta-Analysis (CMA, New Jersey, US) version 2.1. was used to assess effect models. If the p-value was less than 0.10, the random-effect model was used to evaluate heterogeneity. In contrast, a fixed-effect model was used if the P-value > 0.10. Our study's analyses were carried out using Review Manager version 5.3 (RevMan Cochrane, London, UK) and Comprehensive Meta-Analysis (CMA, New Jersey, US) version 2.1.

## Results

### Literature searching

This systematic review and meta-analysis extracted articles from four databases: PubMed, Cochrane, ScienceDirect, and Google Scholar. We found 5536 articles for identification. There was 11 article record removed before screening due to duplication. In the first step screening, there were 5407 articles excluded due to a mismatch of the titles and abstracts. Thus, 120 articles were recorded and continued to the next screening. From 120 articles, the full text was not available for 39 articles. Then, 81 articles were assessed for eligibility according to the inclusion and exclusion criteria and bias quality. There were several articles excluded as follows: no information about duration intervention (n = 16), low sample size (<50 samples) (n = 17), not appropriate study method (n = 12), and insufficient data (n = 29). Finally, seven articles were enrolled in this review (Fig 1).

### Baseline characteristic involves the study and quality assessment

All of the studies were published within the last 20 years, and most were in Asia. The sample sizes of the seven studies ranged from 56 to 131 participants, and the pooled sample size was classified into two groups (PBL vs. conventional learning methods). The participants were in the first year of medical student major (three articles), dentist major (one article), nurse major (two articles), and midwife major (one article). The length of the intervention varied from several months to one year. No specific gender was evaluated (Table 1). All of enrolling studies were in various study types. According to our assessment, two studies had a low risk of bias (score range 9–12 points), and the remaining articles had a medium risk of bias (score range 6–8 points).

### The effectiveness comparison of PBL and conventional learning method

**The critical thinking/knowledge evaluation.** In our finding conventional learning method consist of the conventional method (two articles), LBL (lecture base learning) (two articles), tutorial learning group (one article), and theory-based discussion (one article). Three studies show a higher PBL pre-test score, and three studies show a higher conventional method

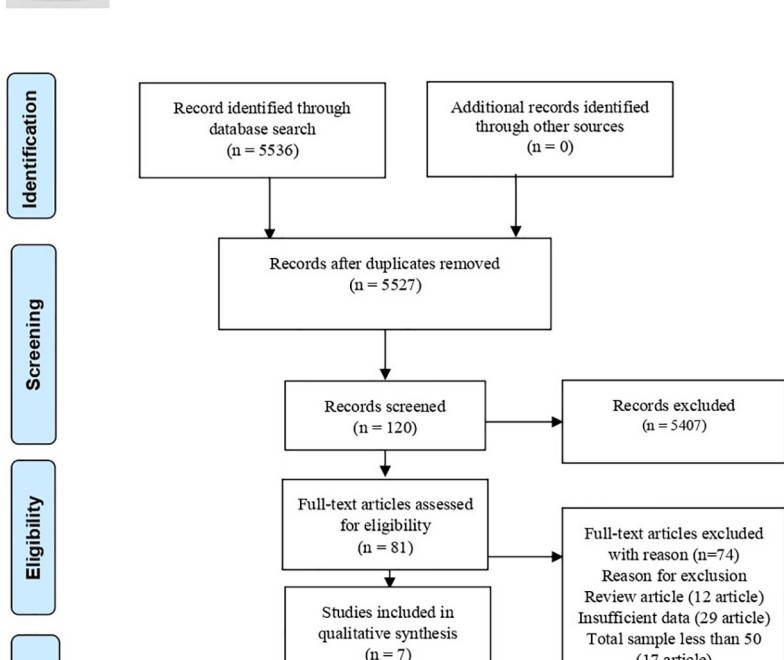

**Fig 1. PRISMA diagram of study selection result.** *From*: Moher D, Liberati A, Tetzlaff J, Altman DG, The PRISMA Group (2009). *Preferred Reporting Items for Systematic Reviews and Meta- Analyses: The PRISMA Statement PLoS Med 6(7): e1000097. doi: 10.1371/journal.pmed1000097 **For more information, visit** www.prisma-statement.org.

**Table 1. Baseline characteristics and quality assessment in each article.**

| Author | Years | Study type | Total sample | Gender | Country | Major | Intervention duration (months) | Age | | Quality assessment |
|---|---|---|---|---|---|---|---|---|---|---|
| | | | | | | | | PBL group | Conventional group | |
| Lohman [13] | 2002 | Case-Control | 74 | M/F | USA | Dentist | 6 | 22.3 ± 3.8 | 21.7 ± 3.8 | Medium risk of bias |
| Tiwari [14] | 2006 | Case-Control | 79 | NA | Hongkong | Nurse | 7 | 20.16 ±1.83 | 20.16 ±1.83 | Low risk of bias |
| Sangestani [15] | 2013 | Randomized quasi-experimental | 56 | M/F | Iran | Midwife | 6 | 18.67 ± 1.71 | 18.57 ± 1.31 | Medium risk of bias |
| Hayashi [16] | 2013 | Randomized cross-matched | 102 | M/F | Japan | Doctor | 12 | NA | NA | Low risk of bias |
| Choi [17] | 2014 | Case-Control | 90 | M/F | Korea | Nurse | 4 | 24.3 ± 2.86 | 24.3 ± 2.86 | Medium risk of bias |
| Tripathi [18] | 2015 | Case-Control | 90 | NA | India | Doctor | 2 | 20.16 ±1.83 | 20.16 ±1.83 | Medium risk of bias |
| Friedrich [19] | 2017 | Case-Control | 131 | M/F | German | Doctor | 12 | 25.64 ±1.92 | 25.82±3.61 | Medium risk of bias |

**Table 2. The outcome of critical thinking/knowledge.** This table provides pre/post-test results in each group and the authors' interpretation of their funding.

| Author | Years | PBL group | | | | Conventional group | | | | Outcome |
|---|---|---|---|---|---|---|---|---|---|---|
| | | Sample (n) | Method | Pre-test score | Post-test score | Sample (n) | Method | Pre-test score | Post-test score | |
| Lohman [13] | 2002 | 37 | PBL | 7.4±1.42 | 9.0 ± 1.41 | 37 | Conventional method | 7.6±1.75 | 8.6 ± 1.38 | Different teaching methods did not have a statistically significant influence on students' knowledge. |
| Tiwari [14] | 2006 | 40 | PBL | 38.03 ± 6.23 | 41.19 ± 5.41 | 39 | LBL | 39.00 ± 4.97 | 40.73 ± 4.56 | There is no significant association between learning methods (PBL vs. lecture-based learning programs) in enhancing critical thinking. |
| Sangestani [15] | 2013 | 22 | PBL | 2.25± 0.99 | 9.64± 0.56 | 34 | LBL | 2.09± 0.99 | 8.24 ± 1.02 | There were no significant differences in the pre-test scores between the control group. |
| Choi [17] | 2014 | 46 | PBL | 51.21 ± 5.61 | 53.41 ± 5.46 | 44 | Conventional method | 56.72 ± 6.16 | 57.54 ± 5.31 | No significant differences were found for technical knowledge from teaching and PBL cases. |
| Tripathi [18] | 2015 | 45 | PBL | 5.2±1.36 | 15.9 ± 2.70 | 45 | Tutorial learning group | 4.9 ± 1.2 | 11.3 ± 1.9 | The study showed a significant finding of knowledge evaluation. The students were evaluated by pre-test and post-test of three modules. |
| Friedrich [19] | 2017 | 76 | PBL | 32.3 ± 17.1 | 32.5 ± 16.9 | 55 | Theory-based discussion | 26.8 ± 16.7 | 30.0 ± 15.6 | The authors established that the difference in mean C-score was not statistically significant for the EG or between the two groups. |

pre-test score. Meanwhile, there was not much difference in the mean score of each group in the pre-test. Evaluation post-test score after the intervention was found to be improved in each group. Post-test scores among PBL groups were mostly higher than the conventional group, except Choi *et al.*, study. It is also in line with Choi *et al.'s* conclusion that stated no significant finding. In addition, Lohman *et al.* study also found that different teaching methods did not significantly influence students' knowledge (Table 2).

**The problem-solving evaluation.** Two articles investigated the critical thinking or knowledge aspect and problem-solving. There was not much difference in the average value of the pre-test and post-test result; meanwhile, Lohman *et al.*; found a significant association between the learning method and problem-solving aspect. Meanwhile, Choi *et al.* did not find a significant difference in each learning method in the problem-solving aspect (Table 3).

**The self-directed learning evaluation.** Three articles evaluated self-directed learning. In the Lohman *et al.* study, the course instructor assessed the student and scored self-directed

**Table 3. Problem-solving score analysis of both groups.**

| Author | Years | PBL group | | | | Conventional group | | | | Outcome |
|---|---|---|---|---|---|---|---|---|---|---|
| | | Sample (n) | Method | Pre-test score | Post-test score | Sample (n) | Method | Pre-test score | Post-test score | |
| Lohman [13] | 2002 | 37 | PBL | 112.15 ± 12.63 | 116.28 ± 15.30 | 37 | Conventional method | 126.95 ± 14.03 | 125.65 ± 17.03 | A significant finding in the problem-solving aspect between PBL group vs. conventional group learning method |
| Choi [17] | 2014 | 46 | PBL | 2.5 ± 1.35 | 7.2 ± 2.11 | 44 | Traditional method | 3.2 ± 1.97 | 6.5 ± 2.57 | No significant difference in comparing the learning method to evaluate the problem-solving aspect |

**Table 4. Self-directed learning score analysis of both groups.**

| Author | Years | PBL group | | | | Conventional group | | | | Outcome |
|--------|-------|-----------|--------|----------------|-----------------|-------------|--------|----------------|-----------------|---------|
| | | Sample (n) | Method | Pre-test score | Post-test score | Sample (n) | Method | Pre-test score | Post-test score | |
| Lohman [13] | 2002 | 37 | PBL | 4.19 ± 0.72 | 4.25 ± 0.74 | 37 | Conventional method | 3.94 ± 0.71 | 4.06 ± 0.86 | This article showed no significant difference in comparing the learning method to evaluate the self-directed learning aspect. |
| Hayashi [16] | 2013 | 51 | PBL | 107.78 ± 12.49 | 110.43 ± 12.05 | 51 | Traditional method | 114.72 ± 12.10 | 113.06 ± 12.64 | The authors performed a powerful association learning method in enhancing self-directed learning. |
| Choi [17] | 2014 | 46 | PBL | 231.6 ± 20.62 | 235.4 ± 20.13 | 44 | Traditional method | 235.3 ± 17.30 | 233.2 ± 21.07 | The difference in learning strategy did not give different self-directed outcomes statistically. |

learning. The higher score obtained, the better level of self-directed learning. Unfortunately, no significant difference was found in comparing learning methods to enhance self-directed learning in all the included studies (Table 4).

## Meta-analysis assessment

Our meta-analysis assessment classified three groups: critical thinking/knowledge, problem-solving, and self-directed learning.

**Critical thinking/knowledge assessment.** Six articles evaluated the critical thinking/knowledge in conventional and PBL groups. This analysis used random effect due to p-value of heterogeneity <0,10. The heterogeneity of these articles was evaluated by using the $I^2$ parameter. According to ReVMan analysis, we established $I^2$ was 93%. It belonged to 75% to 100% classification that had good heterogeneity. We found that for developing critical thinking, PBL was a better program. Unfortunately, there is no significant difference between PBL and conventional learning methods (p = 0.29) (Fig 2). This section had a sub-group analysis according to duration intervention and Asia's critical thinking aspect (Fig 3).

Moreover, the critical thinking studies were regrouped according to the duration of the intervention ($\leq 6\ months\ vs > 6\ months$) and countries (Asia vs. western). The analysis of the duration intervention found no significant difference between PBL and conventional learning methods. Subgroup analysis was assessed by using random effect and fixed-effect models. The

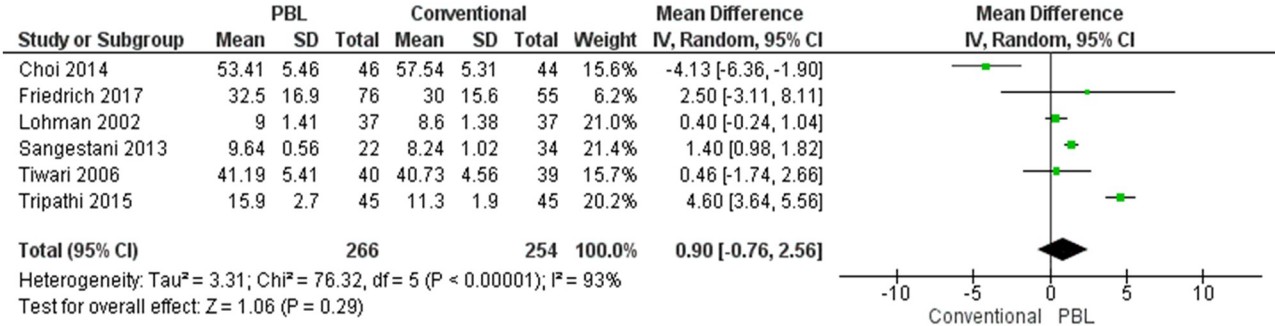

**Fig 2. The analysis of the critical thinking/knowledge aspects.**

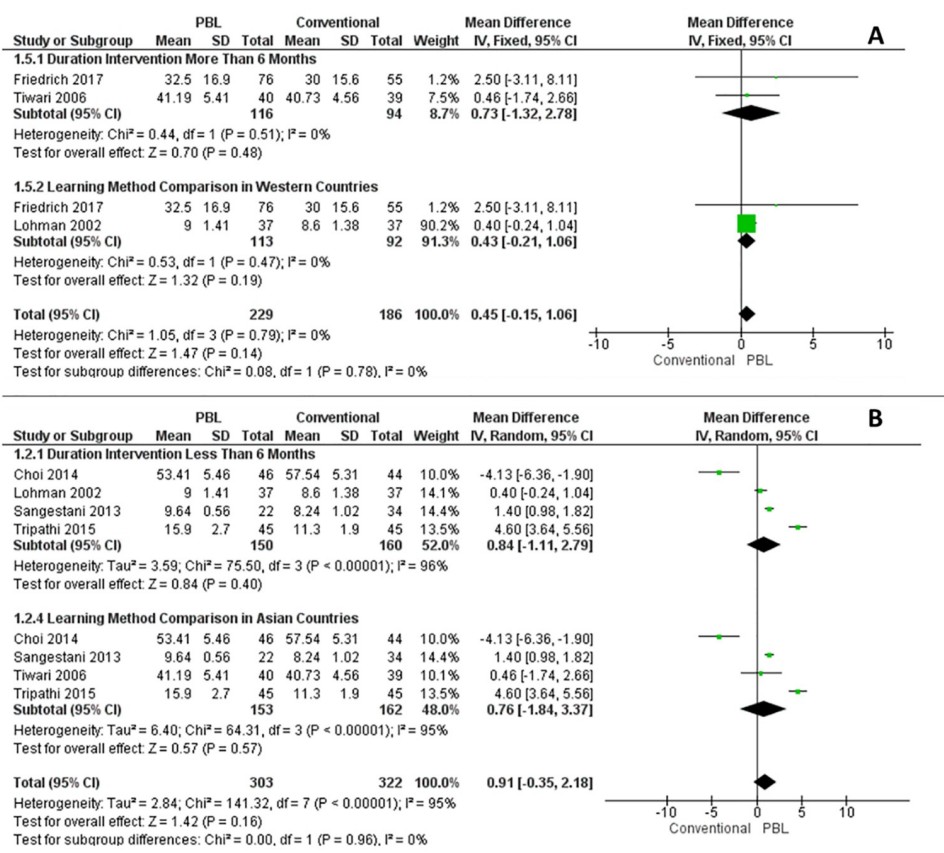

**Fig 3. Subgroup analysis of critical thinking/knowledge.** (A) Fixed effect models. (B) Random effect models.

studies with the duration intervention at more than six months and learning method comparison in western countries subgroup were found to have low heterogeneity (($I^2$ = 0% (might not be important)). However, high heterogeneity scores were found in the studies with duration intervention less than six months ($I^2$ = 96%) and learning method comparison in Asian countries subgroup ($I^2$ = 95%). We discovered no statistical difference between PBL and conventional learning methods in each group even though the test for each subgroup analysis's overall effect from the forest plot graph (diamond) is more inclined to the PBL (Fig 3).

**Problem-solving.** We found two studies that discussed the problem-solving aspect between PBL and conventional learning methods. Both studies had good heterogeneity ($I^2$ = 86% (considerable heterogeneity). The overall results were analyzed by using random effects. It is more toward the conventional teaching for enhancing problem-solving skills, but it was not statistically significant (Fig 4).

**Self-directed learning.** Self-directed learning was evaluated by using a fixed-effect model. The heterogeneity by using the $I^2$ parameter has shown no heterogeneity (0% = might be unimportant)—the overall effect was more inclined toward the conventional method for enhancing self-directed learning. However, there was no statistical difference (p = 0.34) (Fig 5).

## Discussion

Problem Based Learning is a learning method developed to be used as a solution to conventional learning methods that have been used in various disciplines, one of which is health

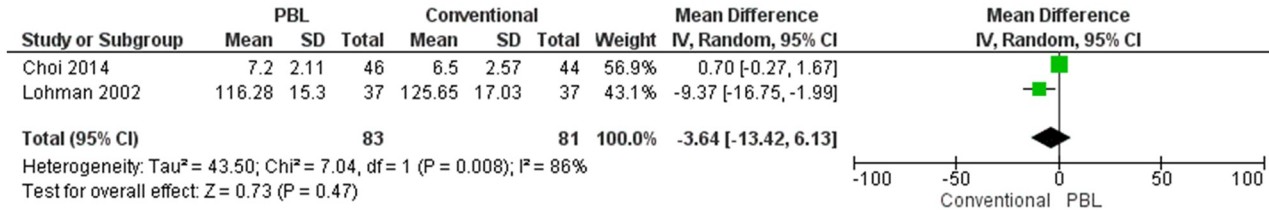

**Fig 4. Evaluation of problem-solving aspect of conventional teaching vs. PBL.**

science. Problem Based Learning is a learning method that emphasizes the active participation of students in solving and solving a given problem, both in group and individual settings, so that it can improve students' skills in analyzing and solving problems [5, 6].

Various studies have been conducted regarding the effectiveness of PBL to be applied in the teaching and learning process [13, 16, 18]. Several factors may influence the implementation of PBL, such as the number of years of study from students, the material taught, and the field of knowledge pursued by students. According to the critical thinking/ knowledge aspect, we found no significant difference between the conventional learning method group and the PBL group (p = 0.29). This finding likely resulted from the lack of association between PBL in enhancing critical thinking/knowledge in the majority of the study. Three studies showed insignificant results from six studies analyzed, and only Tripathi's (2015) [18] has a linear result with our hypothesis. Accordingly, Choi *et al.* stated that their insignificant (p = 0.7) finding was due to a short amount of time of the intervention to produce any meaningful effects [17]. Therefore, intervention duration might not be an absolute factor of PBL effectiveness, as found by Tripathi [18]. Likewise, this study also had the shortest intervention duration but still found significant results. Moreover, research conducted by Li et al. related to critical thinking showed a significant difference between the experimental and control groups (p < 0.001) [20]. Then, Tseng et al., also reported a significant difference in critical thinking scores between the experimental and control groups, where the experimental group had a higher score (p < 0.0001) [21].

The research sample characteristics can also affect the PBL results. In this meta-analysis, we analyzed the medical students' data in their first year. First-year students often experience obstacles in adapting to lecture methods that are different from high school teaching methods [1]. This problem is influenced by various factors, one of which is the difference in lecture methods in each institution. Adaptation to new environments and habits is also a challenge for medical students in the first year. Adaptation to learning methods is a process of response in terms of mental and individual behavior to a demand from the individual or a formal task

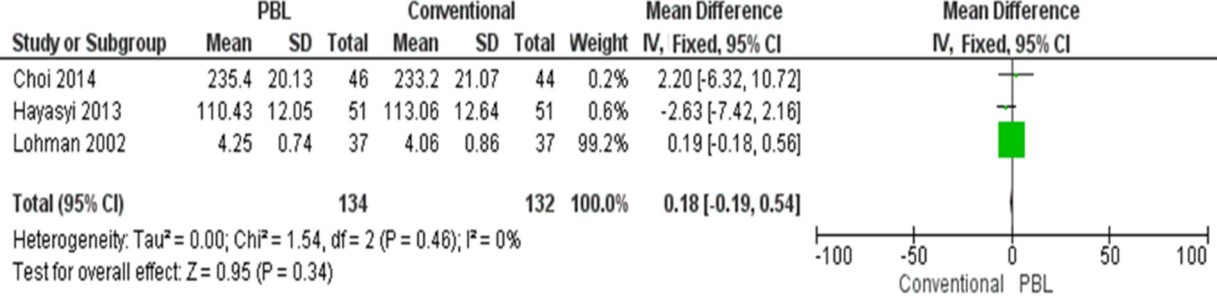

**Fig 5. Self-directed learning evaluation in conventional teaching vs. PBL.**

related to academic work. Therefore, students familiar with the teacher-centered method tend to face difficulty applying the student-centered with PBL method in higher education. They also tend to experience challenges in accepting the study materials, which impact the teaching and learning process in the first semester of lectures for medical students [22]. Those factors explained above could also affect problem-solving and self-directed learning.

Other aspects besides critical thinking/knowledge of the PBL are problem-solving and self-directed learning. We found that PBL is not superior to conventional learning in enhancing problem-solving (p = 0.47). It might be due to the limited studies that assessed this issue and included in this study. The problem-solving aspect was only analyzed in two studies, and they have different results. Choi et al. [17] had a higher total sample, and the study also had a higher weight analysis (56.9%) compared to Lohman et al. [13]. Therefore the results will tend to follow Choi et al. (insignificant finding) [17], besides several aspects as explained above.

Similar results with problem-solving aspect, PBL also failed to show any superiority in increasing self-directed learning compared to the conventional learning method. Two studies in this aspect had shown insignificant results, such as Lohman et al. (2002) [13] and Choi et al., [17]. However, different findings were reported Hayashi et al. (2013) [16]. According to the baseline characteristic of the study, Hayashi's study had a longer duration of intervention than Lohman et al. (2002) [13] and Choi's [17] studies. Thus, it might impact the results because the study subjects were exposed to the intervention much longer, so the desired effect was seen [16]. The PBL learning system that focuses on increasing the active participation of students is expected to be able to improve those aspects compared to using the conventional approach. Research by Tseng et al., 2011 involving 120 nursing students (51 in the experimental group, 69 in the control group) showed a significant difference in self-directed learning scores, where the experimental group had a higher mean value than the control group (p < 0.0001) [21]. Three aspects of PBL were evaluated in this meta-analysis, and none were significant. Unfortunately, the specific aspect that might impact the result did not mention or explained in each study in detail.

The problem-based learning method has been used widely, and to the best of our knowledge, further investigation about this learning method is needed. The strength of this study was that our meta-analysis evaluated the specific outcome of PBL such as critical thinking/knowledge assessment, problem-solving, and self-directed learning. Several studies discuss the PBL effect on general learning outcomes and specific backgrounds [9, 14, 18, 19, 23]. Our meta-analysis not only provided pre-test and post-test scores in each group, but we also explained the outcome in each study. Furthermore, we noted that high levels of heterogeneity across studies were found in this meta-analysis. Factors that may cause heterogeneity include the sample from different countries with different backgrounds. Second, the instrument used to evaluate the PBL progression in each study was different. Third, the duration of intervention was also varied, bringing different outcomes. All of these factors may contribute to our meta-analysis heterogeneity. Subgroup analysis has been conducted to minimize the heterogeneity. This method can only reduce the heterogeneity in terms of the critical thinking/acknowledgment aspect, especially when the duration of intervention was more than six months and when the learning method was compared in the Western country sub-group. Meanwhile, no effect was found in terms of heterogeneity when duration of intervention was less than six months, and the learning method was conducted in the Asian countries sub-group. It might be due to several factors that have been pointed out above. Unfortunately, we cannot run subgroup analyses due to limited studies discussing this topic.

Additionally, we believe that further primary study is needed to evaluate the effectiveness of PBL. A multicenter approach is suggested as the most appropriate method to identify the cumulative effect and the difference between geographic areas or races. Moreover, researchers

can also compare between educational centers as well as the impact of culture and technological progress of the local area in the implementation of PBL due to the rarity of the study regarding these topics. Psychological aspects also need to be discussed because medical students in the first year may still have the learning method from high school, potentially affecting the PBL.

## Conclusion

In conclusion, according to our analysis, PBL is not superior to conventional learning in improving critical thinking/knowledge, problem-solving and self-directed learning in first-year medical students. In addition, our meta-analysis had several limitations, such as only evaluating the learning outcomes in the first year, and no studies were found with multiyear approach. We could not equate the instruments used in PBL and did not evaluate specifically based on the study program. We also could not assess the socio-demography that might contribute to their learning process, particularly their social culture. Therefore, a multicenter approach is suggested as the most appropriate method to identify the cumulative effect and the difference between geographic areas or races.

## Supporting information

**S1 File.**
(RAR)

## Author Contributions

**Conceptualization:** Ida Bagus Amertha Putra Manuaba, Yi -No, Chien-Chih Wu.

**Data curation:** Ida Bagus Amertha Putra Manuaba.

**Formal analysis:** Ida Bagus Amertha Putra Manuaba, Yi -No.

**Investigation:** Ida Bagus Amertha Putra Manuaba.

**Methodology:** Ida Bagus Amertha Putra Manuaba, Yi -No, Chien-Chih Wu.

**Supervision:** Yi -No, Chien-Chih Wu.

**Validation:** Yi -No, Chien-Chih Wu.

**Writing – original draft:** Ida Bagus Amertha Putra Manuaba.

**Writing – review & editing:** Ida Bagus Amertha Putra Manuaba, Yi -No, Chien-Chih Wu.

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
