## [Decision Letter · Decision Letter 0]

20 Jan 2022

PONE-D-21-22466The effectiveness of problem-based learning in the education of first-year medical students in multidisciplinary aspects: a meta-AnalysisPLOS ONE

Dear Dr. Wu,

Thank you for submitting your manuscript to PLOS ONE. After careful consideration, we feel that it has merit but does not fully meet PLOS ONE’s publication criteria as it currently stands. Therefore, we invite you to submit a revised version of the manuscript that addresses the points raised during the review process.

We look forward to receiving your revised manuscript.

Kind regards,

Rohit Kunnath Menon

Academic Editor

PLOS ONE

Journal Requirements:

Reviewers' comments:

Reviewer's Responses to Questions

**Comments to the Author**

1. Is the manuscript technically sound, and do the data support the conclusions?

Reviewer #1: Yes

Reviewer #2: Partly

2. Has the statistical analysis been performed appropriately and rigorously? 

Reviewer #1: Yes

Reviewer #2: Yes

3. Have the authors made all data underlying the findings in their manuscript fully available?

Reviewer #1: Yes

Reviewer #2: Yes

4. Is the manuscript presented in an intelligible fashion and written in standard English?

Reviewer #1: No

Reviewer #2: Yes

5. Review Comments to the Author

Reviewer #1: The study titles as ‘The effectiveness of problem-based learning in the education of first-year medical students in multidisciplinary aspects: a meta-Analysis. However, the objective is to synthesize the effectiveness of PBL in improving knowledge/critical thinking, problem-solving and self-directed learning in first-year medical students. The title didn’t reflect the study exactly. Hence, a revision is suggested. Also the term multidisciplinary approach may lead mislead the readers.

Abstract.At this point of time, Problem based learning (PBL) is not an innovative method as it is used widely used as a constructivist pedagogic philosophy and an instructional format for promoting contextual, co-operative and self-directed learning. Moreover, PBL is used not only for medical education. Please revise the statement.

The authors are encouraged to discuss the challenges faced by the first year medical students in adapting PBL to justify the study objective. The reasons such as current studies focused on students in advanced years and lack of such studies in the first-year students should not be the only reason for this study.

Introduction: The authors’ statement on the effectiveness of PBL “Although PBL-used widely, still there is no substantial evidence to support the claim that PBL is proven to be more effective than conventional lecture methods” merely highlighting the comparativeness between PBL and Lecture. In fact, there are reports on the effectiveness of PBL, which improves student engagement by enabling knowledge and information sharing and discussion. The delivery of curriculum in medical education has a variety of methods to teach. It would be appropriate if authors focus more on how PBL is compared with other forms of teaching as well.

The last paragraph of introduction, explained the results of 2 studies [5,6] and mentioned that there has been no meta-analysis that discusses the effectiveness of PBL in first-year health students in multidisciplinary aspects. What does it mean by ‘health students’.

This paragraph need revision. Highlighting the connection between the research gap and the study objective clearly.

The purpose of this meta-analysis is to synthesize the evidence on the effectiveness of PBL. A revision is required in that statement mentioned about the purpose as the meta-analysis is to examine an effect within a collection of studies.

Methodology: Eligibility criteria and data extraction is not clear.

Under quality assessment, it is mentioned If the conflict has not been settled, a Senior Researcher consultation has been carried out. Please provide the detail of the reviewer.

Reference 10 is not a RCT- it is a Quasi-experimental research, which specifically lacks the element of random assignment to treatment or control. Reference 11 also a randomised cross-matched study. Please review accordingly.

Discussion: The Discussion section is an important scientific component in a manuscript describing a meta-analysis, as the authors should discuss their current findings in the context of the available scientific literature and existing knowledge. The readers would expect possible reasons for the positive or negative results of the meta-analysis. As meta-analyses are usually synthesizing the existing evidence from multiple primary studies, authors can recommend key suggestions for conducting and/or reporting future primary studies.

Conclusion:The PBL method in the learning process is a breakthrough made to replace the conventional learning system, which has many weaknesses. The statement is not the conclusion from the study. The authors may discuss such points in the discussion. There is a lack of connection between the research question and conclusion. It is suggested to review the conclusion.

Also, there is no limitation is mentioned. Provide the list of limitations of this study.

Reviewer #2: This meta-analysis paper aimed to synthesize the effectiveness of PBL in improving knowledge/critical thinking, problem-solving and self-directed learning in first-year medical students

However few points need to be addressed

1. Just two databases is often not acceptable for data search.

Minimum three is required to make sure you haven't missed any publication.

Author needs to use Scopus or any other database and do the search as well as update Prisma chart

2. Heterogeneity is very high in almost all forest plots. These needs to be explored and discussed more in detail

3. Authors need to critically evaluate the included articles in their discussion. The discussion is too superficial.

4. Few errors in grammar could be identified

6. PLOS authors have the option to publish the peer review history of their article (what does this mean?). If published, this will include your full peer review and any attached files.

Reviewer #1: No

Reviewer #2: No

---

## [Author Response · Author response to Decision Letter 0]

6 Apr 2022

Reviewer #1

1. Title

The study titles as 'The effectiveness of problem-based learning in the education of first-year medical students in multidisciplinary aspects: a meta-Analysis. However, the objective is to synthesize the effectiveness of PBL in improving knowledge/critical thinking, problem-solving and self-directed learning in first-year medical students. The title didn't reflect the study exactly. Hence, a revision is suggested. Also the term multidisciplinary approach may lead mislead the readers 

The word “multidisciplinary” has been removed from the title because there was no discussion regarding the differences in the effectiveness of PBL in each medical study program

Title revision:

“The effectiveness of Problem Based Learning in improving critical thinking, problem-solving and self-directed learning in first-year medical students: a meta-Analysis”

2. Abstract

At this point of time, Problem based learning (PBL) is not an innovative method as it is used widely used as a constructivist pedagogic philosophy and an instructional format for promoting contextual, co-operative and self-directed learning. Moreover, PBL is used not only for medical education. Please revise the statement. The authors are encouraged to discuss the challenges faced by the first year medical students in adapting PBL to justify the study objective. The reasons such as current studies focused on students in advanced years and lack of such studies in the first-year students should not be the only reason for this study.

The author has revised the statement that “PBL is an innovative method”. The author focuses on assessing the advantages of PBL compared to conventional learning methods for first-year medical students due to the unclear effect of this learning method in this population.

3. Introduction

- The authors' statement on the effectiveness of PBL "Although PBL-used widely, still there is no substantial evidence to support the claim that PBL is proven to be more effective than conventional lecture methods" merely highlighting the comparativeness between PBL and Lecture. In fact, there are reports on the effectiveness of PBL, which improves student engagement by enabling knowledge and information sharing and discussion. The delivery of curriculum in medical education has a variety of methods to teach. It would be appropriate if authors focus more on how PBL is compared with other forms of teaching as well. 

The author has added a sentence that focuses more on the effectiveness of PBL compared to other forms of teaching methods (especially conventional learning methods, which are used as comparisons in this study).

- The last paragraph of introduction, explained the results of 2 studies [5,6] and mentioned that there has been no meta-analysis that discusses the effectiveness of PBL in first-year health students in multidisciplinary aspects. What does it mean by health students? �

The health student that the author means is a medical student. The medical student is a student enrolled at a medical school, including a doctor, dentist, nurse, physiotherapist, or midwife. If there is any difference in our perspective, please let us know so we can revised the terms in this manuscript.

- This paragraph need revision. Highlighting the connection between the research gap and the study objective clearly. The purpose of this meta-analysis is to synthesize the evidence on the effectiveness of PBL. A revision is required in that statement mentioned about the purpose as the meta-analysis is to examine an effect within a collection of studies.

The author has revised the study objectives according to the research gap

4. Methodology

- Eligibility criteria and data extraction is not clear.

- We have adjusted the eligibility criteria and data extraction to make it clearer. Therefore, the eligibility criteria and data extraction were as follows:

- The inclusion criteria for this study included: (1) research subjects were medical students at the first year (first or second semester), (2) study that evaluated the knowledge/critical thinking, problem-solving and self-directed learning of the student, (3) studies that providing sufficient data for calculation of mean difference and 95%CI, p-value, and study heterogeneity. Meanwhile, the exclusion criteria were as follows: (1) studies with insufficient data, (2) samples less than 50, (3) intervention less than 1 year, (4) review, letter to the editor, and comments articles. Data extraction was conducted by two authors (I.BA.P.M, Y.N). Both authors independently screened the collected article's title, abstract, and full text. Two reviewers extracted the data was then extracted to Google Spreadsheet by two reviewers (I.BA.P.M, Y.N). Information was derived from each article included in this study as follows: (1) first author's name and year of release, (2) age of the participant, (3) interventional and control method, (4) sample cases and control sizes, (5) country of study, (6) study program, (7) duration of PBL intervention, (8) score of PBL and control method. Two independent authors carried out data extraction to prevent human mistakes. If there is a disagreement, a discussion will be held to discuss the solution.

- Under quality assessment, it is mentioned If the conflict has not been settled, a Senior Researcher consultation has been carried out. Please provide the detail of the reviewer.

- We have clarified the roles of 2 authors who act as reviewers in assessing the quality of each article screened in this study. The quality assessment section that we have revised is as follows:

- The quality of the studies was assessed by two independent authors (I.BA.P.M, Y.N) to ensure each sample's validity and prevent the possible exaggeration of each study. The authors use major and minor criteria in assessing the risk of bias for quality assessment. There were 4 major and 4 minor criteria. The authors assigned 2 points each to the major criteria and 1 point each to the minor criteria, so the total score could be 12 points. If the article got 9-12 points, then it assigned as “low-risk bias”, if the article got 6-8 points, then it assigned as medium risk bias”, and if the article got < 5 points, then it assigned as “high risk of bias”. When there was a disagreement between two separate authors, a discussion was held. If the conflict has not been settled, the two authors discuss it with the third author (C.C.W).

- Reference 10 is not a RCT- it is a Quasi-experimental research, which specifically lacks the element of random assignment to treatment or control. Reference 11 also a randomised cross-matched study. Please review accordingly.

- Reference no 10 (Sangestani et al) � now become reference no 13 � this study type has been adjusted to a randomized quasi-experimental study. Based on that article, the design was quasi-experimental, but randomization was carried out to determine the treatment group (PBL group)

- Reference no 11 (Hayashi et al) � now become reference no 14 � this study type has been adjusted to a randomized cross-matched study.

5. Discussion

- The Discussion section is an important scientific component in a manuscript describing a meta-analysis, as the authors should discuss their current findings in the context of the available scientific literature and existing knowledge. The readers would expect possible reasons for the positive or negative results of the meta-analysis. As meta-analyses are usually synthesizing the existing evidence from multiple primary studies, authors can recommend key suggestions for conducting and/or reporting future primary studies.

Positive and negative results are discussed in the discussion. We highlight the revision with green color in the discussion. Suggestion for the further primary study was discussed in the last paragraph of the discussion (green highlight). Here we attached our suggestion below :

“……….In addition, primary research is still needed to evaluate the effectiveness of PBL. We suggest conducting the primary study about PBL by carrying it out at several different educational centers so that in addition to getting the results of the PBL learning evaluation, researchers can also compare between educational centers. Researchers can also evaluate aspects of the culture and technological progress of the local area that can be a supporting or hindering factor in the implementation of PBL. Because very rarely this aspect is raised in PBL research. Psychological aspects also need to be discussed, because medical students in the first year may still be adopted to the learning obtained so that it affects the PBL implementation process.”

6. Conclusion

The PBL method in the learning process is a breakthrough made to replace the conventional learning system, which has many weaknesses. The statement is not the conclusion from the study. The authors may discuss such points in the discussion. There is a lack of connection between the research question and conclusion. It is suggested to review the conclusion. We revised the conclusion according to the suggestion, here we attached the revision :

“In conclusion, according to our finding PBL learning method do not effective in improving critical thinking/knowledge, problem-solving and self-directed learning in first-year medical students. In addition, our meta-analysis had several limitations, such as only evaluating the learning outcomes in the first year and does not evaluate continuously in the following year. We could not equate the instruments used in PBL learning and did not evaluate specifically based on the study program. We could not assess the socio-demography that might contribute to their learning process, particularly in their social culture.” 

- Also, there is no limitation is mentioned. Provide the list of limitations of this study.

The author has provided the limitation in the discussion. Meanwhile, according to this comment, we moved the study limitation to the conclusion

Reviewer #2

This meta-analysis paper aimed to synthesize the effectiveness of PBL in improving knowledge/critical thinking, problem-solving and self-directed learning in first-year medical students. However few points need to be addressed

1. Just two databases is often not acceptable for data search. Minimum three is required to make sure you haven't missed any publication. Author needs to use Scopus or any other database and do the search as well as update Prisma chart

The author has used more than two databases, namely Pubmed, ScienceDirect, Cochrane and Google Scholar.

2. Heterogeneity is very high in almost all forest plots. These needs to be explored and discussed more in detail

We have revised it and signed it with a green highlight

3. Authors need to critically evaluate the included articles in their discussion. The discussion is too superficial.

We have revised it and signed it with a green highlight

4. Few errors in grammar could be identified

The author has revised the grammar of this manuscript

---

## [Decision Letter · Decision Letter 1]

7 Jun 2022

PONE-D-21-22466R1The Effectiveness of Problem Based Learning in Improving Critical Thinking, Problem-Solving and Self-Directed Learning in First-Year Medical Students: A Meta-AnalysisPLOS ONE

Dear Dr. Wu,

Thank you for submitting your manuscript to PLOS ONE. After careful consideration, we feel that it has merit but does not fully meet PLOS ONE’s publication criteria as it currently stands. Therefore, we invite you to submit a revised version of the manuscript that addresses the points raised during the review process.

Kindly make the suggested minor revisions.

We look forward to receiving your revised manuscript.

Kind regards,

Rohit Kunnath Menon

Academic Editor

PLOS ONE

Journal Requirements:

Reviewers' comments:

Reviewer's Responses to Questions

**Comments to the Author**

1. If the authors have adequately addressed your comments raised in a previous round of review and you feel that this manuscript is now acceptable for publication, you may indicate that here to bypass the “Comments to the Author” section, enter your conflict of interest statement in the “Confidential to Editor” section, and submit your "Accept" recommendation.

Reviewer #1: All comments have been addressed

Reviewer #2: All comments have been addressed

2. Is the manuscript technically sound, and do the data support the conclusions?

Reviewer #1: Yes

Reviewer #2: Yes

3. Has the statistical analysis been performed appropriately and rigorously? 

Reviewer #1: N/A

Reviewer #2: Yes

4. Have the authors made all data underlying the findings in their manuscript fully available?

Reviewer #1: Yes

Reviewer #2: Yes

5. Is the manuscript presented in an intelligible fashion and written in standard English?

Reviewer #1: No

Reviewer #2: Yes

6. Review Comments to the Author

Reviewer #1: Introduction The adaptation process for first-year medical students is an important problem because it is one of the factors that significantly affect educational outcomes. reference ?

A study on the adaptation process of first-year medical students involving 200 participants showed that 63% of students had problems adapting to the learning process. reference?

Problem-based learning (PBL) is an educational method that emphasizes active participation, problem-solving, and critical thinking skills compared to conventional medical education practices.[3] This statement is contradictory to the results. Ideally authors can consider to keep this point the later stage, mostly at the end of the paragraph rather than starting with this.

Discussion

Various studies have been conducted regarding the effectiveness of PBL to be applied in

274 the teaching and learning process. - References

Meanwhile, intervention duration might not be an absolute factor of PBL effectiveness because Tripathi [16] found significant results, and this study had the shortest intervention duration compared to other studies. This is statement is not clear.

In this meta-analysis, we evaluated medical students in their first year. - Actually the authors evaluated the data only. Please revise

First-year students often experience obstacles is adapting to lecture methods that are different from high school teaching methods. reference?

Several studies discuss the PBL effect on general learning outcomes and specific backgrounds. What are the studies ? references.

Under conclusion section, authors mentioned that "In conclusion, according to our findings",-doesn't sound logical. Pls consider to revise.

Reviewer #2: The authors have addressed all the comments provided by reviewer. The paper can be accepted and published

7. PLOS authors have the option to publish the peer review history of their article (what does this mean?). If published, this will include your full peer review and any attached files.

Reviewer #1: **Yes: **MariKannan Maharajan

Reviewer #2: No

---

## [Author Response · Author response to Decision Letter 1]

22 Aug 2022

Dear Editors and Reviewers,

Thank you for your corrections to our manuscript. We have made several revisions according to the reviewer’s comments and we hope that this manuscript is now met your standard. The revisions are as follows: 

Introduction 

1. The adaptation process for first-year medical students is an important problem because it is one of the factors that significantly affect educational outcomes. reference ? 

Response: We have put the reference in the text.

2. A study on the adaptation process of first-year medical students involving 200 participants showed that 63% of students had problems adapting to the learning process. reference? 

Response: We have put the reference in the text.

3. Problem-based learning (PBL) is an educational method that emphasizes active participation, problem-solving, and critical thinking skills compared to conventional medical education practices.[3] This statement is contradictory to the results. Ideally authors can consider to keep this point the later stage, mostly at the end of the paragraph rather than starting with this.

Response: We have revised this section and moved it to the end of paragraph.

Discussion

4. Various studies have been conducted regarding the effectiveness of PBL to be applied in

274 the teaching and learning process. – References?

Response: We have put the references

5. Meanwhile, intervention duration might not be an absolute factor of PBL effectiveness because Tripathi [16] found significant results, and this study had the shortest intervention duration compared to other studies. This is statement is not clear.

Response: We have revised the statements

6. In this meta-analysis, we evaluated medical students in their first year. - Actually the authors evaluated the data only. Please revise

Response: We have revised the statement

7. First-year students often experience obstacles is adapting to lecture methods that are different from high school teaching methods. reference?

Response: We have put the reference

8. Several studies discuss the PBL effect on general learning outcomes and specific backgrounds. What are the studies ? references 

Response: We have put the references

9. Under conclusion section, authors mentioned that "In conclusion, according to our findings",-doesn't sound logical. Pls consider to revise.

Response: We have revised the statement

Best Regards,

Chien-Chieh Wu

---

## [Editor Report · Decision Letter 2]

26 Oct 2022

The Effectiveness of Problem Based Learning in Improving Critical Thinking, Problem-Solving and Self-Directed Learning in First-Year Medical Students: A Meta-Analysis

PONE-D-21-22466R2

Dear Dr. Wu,

We’re pleased to inform you that your manuscript has been judged scientifically suitable for publication and will be formally accepted for publication once it meets all outstanding technical requirements.

Kind regards,

Huijuan Cao, Ph.D.

Academic Editor

PLOS ONE

Additional Editor Comments (optional):

With all the revisions you've made responding to reviewers' comments, I have just one more suggestion. Please clarify the grouping method of all the included studies. Is there any study used cluster randomization method? If so, the pooling analysis with data from both cluster randomized trial and individual randomized trial is different to the commone one.